# GWAS-Based Identification of New Loci for Milk Yield, Fat, and Protein in Holstein Cattle

**DOI:** 10.3390/ani10112048

**Published:** 2020-11-05

**Authors:** Liyuan Liu, Jinghang Zhou, Chunpeng James Chen, Juan Zhang, Wan Wen, Jia Tian, Zhiwu Zhang, Yaling Gu

**Affiliations:** 1School of Agriculture, Ningxia University, Yinchuan 750021, Ningxia, China; lly000917@gmail.com (L.L.); zjhang027@gmail.com (J.Z.); zhangjkathy@126.com (J.Z.); 2Department of Crop and Soil Sciences, Washington State University, Pullman, Washington, DC 99164, USA; chun-peng.chen@wsu.edu; 3Animal Husbandry Workstation, Yinchuan 750001, Ningxia, China; 13709503296@163.com (W.W.); tianjia214@126.com (J.T.)

**Keywords:** dairy, milk production, quality traits, *DGAT1*, FarmCPU

## Abstract

**Simple Summary:**

Understanding the genetic architecture underlying milk production traits in cattle is beneficial so that genetic variants can be targeted toward the genetic improvement. In this study, we performed a genome-wide association study for milk production and quality traits in Holstein cattle. In the total of ten significant single-nucleotide polymorphisms (SNPs) associated with milk fat and protein, six are located in previously reported quantitative traits locus (QTL) regions. The study not only identified the effect of *DGAT1* gene on milk fat and protein but also found several novel candidate genes. In addition, some pleiotropic SNPs and QTLs were identified that associated with more than two traits, these results could provide some basis for molecular breeding in dairy cattle.

**Abstract:**

High-yield and high-quality of milk are the primary goals of dairy production. Understanding the genetic architecture underlying these milk-related traits is beneficial so that genetic variants can be targeted toward the genetic improvement. In this study, we measured five milk production and quality traits in Holstein cattle population from China. These traits included milk yield, fat, and protein. We used the estimated breeding values as dependent variables to conduct the genome-wide association studies (GWAS). Breeding values were estimated through pedigree relationships by using a linear mixed model. Genotyping was carried out on the individuals with phenotypes by using the Illumina BovineSNP150 BeadChip. The association analyses were conducted by using the fixed and random model Circulating Probability Unification (FarmCPU) method. A total of ten single-nucleotide polymorphisms (SNPs) were detected above the genome-wide significant threshold (*p* < 4.0 × 10^−7^), including six located in previously reported quantitative traits locus (QTL) regions. We found eight candidate genes within distances of 120 kb upstream or downstream to the associated SNPs. The study not only identified the effect of *DGAT1* gene on milk fat and protein, but also discovered novel genetic loci and candidate genes related to milk traits. These novel genetic loci would be an important basis for molecular breeding in dairy cattle.

## 1. Introduction

Milk production and quality are the most important economic traits in the dairy industry. Most milk phenotypes are quantitative traits that often controlled by both environmental factors and multiple genes. A large number of studies have revealed numerous quantitative traits locus (QTL) regions for milk-related traits in dairy cattle population around the world over the past 20 years (CatttleQTLdb: https://www.animalgenome.org/cgi-bin/QTLdb/BT/index), and many researchers conducted meta-analysis to identify genetic variants based on GWAS results for milk-related traits in different cattle breeds [1,2,3]. Whereas, in Chinese Holstein population, a previous study used 50,000 single-nucleotide polymorphisms (SNPs) and revealed some SNPs associated with milk production traits [4]. The aim of present study was to find new genetic loci in this population by using a higher density marker information.

Research has shown that high-density genotype could provide markers close to the QTL and help in fine mapping of causative mutations [5]. Vanraden et al. reported that high-density marker increased the precision of QTL detection in cattle population [6]. In addition, a study reported that the genomic prediction accuracy increased when the marker density was increased in cattle [7]; therefore, it is necessary to use dense genotype to identify important genetic variation, and provide some useful information for molecular breeding of dairy cattle and understanding the genetic architecture of milk traits.

Genome-wide association studies (GWAS) are very helpful for further genomic selection (GS) as they have proven to be a powerful method for identifying potential genetic variants, especially single-nucleotide polymorphisms (SNPs) associated with complex traits in humans and animals [8,9,10]. Resende et al. suggested that the prediction accuracy reached maximum when the genomic relationship matrix was constructed using causative quantities trait nucleotides (QTNs) [11]. Incorporating significant markers from GWAS results can improve the prediction accuracy in dairy cattle [1,12,13,14].

In this study, we conducted GWAS using Illumina BovineSNP150 BeadChip which contains about 150,000 SNPs. The population was from Holstein cows raised in the Ningxia area of northwest China. Our objectives were to identify new genetic variants associated with five milk production and quality traits, including milk yield, fat and protein. We expect the newly identified genetic variants and potential candidate genes would become valuable resources for genetic evaluation.

## 2. Material and Methods

### 2.1. Population and Phenotypic Data

The studied population was Holstein cows that were raised on 22 dairy farms in the Ningxia area of China. In total, about 452,920 test-day records estimated breeding values from 61,600 cows spanning a 9-year period (2011–2019) at their first lactation. The estimated breeding values (EBVs) as phenotypes to implement association analysis, milk yield (MY), fat yield (FY), protein yield (PY), fat percentage (FP), and protein percentage (PP) measurements were recorded once a month for each cow after calving. The milk yield was automatically recorded by the milking system on each farm, the milk components are tested by the Dairy Herd Improvement lab at Animal Husbandry of Extension Station in Ningxia, using spectrometers. FY was calculated as (FP*MY)/100; PY was calculated as (PP*MY)/100. The distribution of phenotypes and correlations between the different phenotypic traits are illustrated in Appendix A.

### 2.2. Estimated Breeding Values

The Derivative-free approach to MUltivariate analysis (DMU) package [15] was used to estimate breeding values using Random Regression Test-Day Model [16,17]. We considered herd-test day and calving year-season as fixed effects, calving month-age as fixed regression effect, and individual additive genetic effect and permanent environment effect as random regression effects. Both fixed and random regressions were modeled using a 4th order Legendre polynomial [18]. The model equation is as follows:(1)$$y_{ijklm}=\;HTD_{i}+cays_{j}+\overset{4}{\underset{m=0}{\sum }}b_{km}X_{m}\left(\omega \right)+\overset{4}{\underset{m=0}{\sum }}a_{lm}X_{m}\left(\omega \right)+\overset{4}{\underset{m=0}{\sum }}p_{lm}X_{m}\left(\omega \right)+e_{ijklm}$$
where $y_{ijklm\;}$ is the test-day records; $HTD_{i\;}$
is the fixed effect of the *i*th herd-test day (*i* = 1, …, 1913); $cays_{j\;}$
is the fixed effect of the *j*th calving year-season (*j* = 1, …, 36); $b_{km\;}$ is fixed regression coefficient for the *k*th class of calving month-age (*k* = 1, …, 8); $a_{lm\;}$
is random regression coefficient for additive genetic effects specific to cow *l*; $p_{lm\;}$
is random regression coefficient for permanent environment effects specific to cow *l*; $X_{m}\left(\omega \right)\;$
is the *m*th covariate of Legendre polynomial; ω is the days of lactation after standardization; and $e_{ijklm}$ is the random residual effects, hypothesizing the residuals are homogeneous. The variance–covariance matrix is as follows [19]:(2)$$Var\left[\begin{array}{c} a\\ p\\ e \end{array}\right]=\;\left[\begin{array}{ccc} G⊗A & 0 & 0\\ 0 & I⊗P & 0\\ 0 & 0 & R \end{array}\right]$$
where a is additive genetic random regression coefficient vector; *p* is permanent environment random regression coefficient vector; *G* is variance–covariance matrix of additive genetic random regression coefficient; *A* is numerator relationship matrix; *p* is variance–covariance matrix of permanent environment random regression coefficient; *I* is the identity matrix; *R* is diagonal matrix of residual variance (*I*$\sigma_{e}^{2}$), which hypothesizes the residuals are homogeneous. The homogeneous option dramatically reduces computing time without sacrifice as there is a minimal difference between the homogeneous model and the heterogenous model.

A heatmap of estimated breeding values for milk production traits is illustrated in Appendix A.

### 2.3. Genotypic Data

Blood samples from the 1220 cows were collected by cattle farm staff in this study. DNA was extracted and genotyping was carried out by Compass Biotechnology (http://www.kangpusen.com/) using the Illumina BovineSNP150 BeadChip. Bos_taurus_UMD_3.1 as the genome reference. In total, there were 124,743 variants for the association analysis after conducting quality control using Plink software [20]. Markers were removed if (1) the call rate of an individual genotype was less than 95%, (2) the call rate of a single SNP genotype was less than 90%, and (3) if the minor allele frequency (MAF) of an SNP was less than 0.05 and deviated from Hardy–Weinberg equilibrium (*p* < 1.0 × 10^−6^). We calculated marker intervals and linkage disequilibrium (LD) to estimate R square for all markers and plotted the marker distribution as show in Figure 1.

### 2.4. Principal Component Analysis

Principal component analysis (PCA) was conducted using R function Prcomp() on 1220 cows genotyped with 124,743 markers covering the whole genome to study the population structure [21,22]. Most of these 1220 cows were the progeny of the frozen semen imported from multiple countries.

### 2.5. Association Analysis

We performed association analysis by a multi-locus linear mixed model using the FarmCPU (Fixed and random model Circuitous Probability Unification) [23]. FarmCPU method implements marker tests with associated markers as covariates in a fixed effect model and optimization on the associated covariate markers in a random effect model separately [23]. As we know, population stratification is an important factor that can cause false positive in association studies [22]. Therefore, the present study fitted the first three principal components (PCs) as covariate variables in the GWAS models [23], the fixed effect model is as follows:(3)$$y=Xb_{X}+M_{t}b_{t}+S_{j}d_{j}+e$$
where y is the EBVs of individual; X is a matrix of fixed effect for the first three PCs; $M_{t}$ is the genotype matrix of t pseudo Quantitative Traits Nucleotides (QTNs),initiated as an empty set; $b_{X}$ and $b_{t}$ are the corresponding effects of X and $M_{t},$ respectively; $S_{j}$ is the genotype of the j marker; $d_{j}$ is the corresponding effect; e is the vector of residuals $e\;\sim \;N\left(0,I\sigma_{e}^{2}\right)$. The random effect model is as follows:(4)y= u+e
where y and e stay the same as in the fixed effect model; u is the genetic effect of the individual and $u\;\sim \;N\left(0,K\sigma_{u}^{2}\right)$, in which K is the kinship matrix derived from the pseudo QTNs.

The genome wise threshold corresponding type I error of 1% was 4.0 × 10^−7^ after Bonferroni multiple test correction (5%/124,743).

### 2.6. Annotation of Candidate Gene and Pathway Analysis

The genome reference Bos_taurus_UMD_3.1 was used to search candidate gene. The average pairwise LD was 0.46 corresponding to adjacent marker distance of 120 kb. This range was used to search candidate genes. The online websites “https://oct2018.archive.ensembl.org/Bos_taurus/Info/Index”, “https://www.ncbi.nlm.nih.gov/gene/”, https://www.genome.jp/kegg/pathway.html, https://david.ncifcrf.gov/home.jsp were used for functional analysis and pathway analysis of the candidate genes by GWAS.

## 3. Results

### 3.1. Phenotypic and Estimated Genetic Parameters

Phenotype distributions and correlations among phenotypic traits, estimated breeding values, and residuals are shown in the Appendix A. There were strong positive phenotypic correlations between “yield” type of traits, including MY, PY, and FY. Their phenotypic correlations were 0.90 (MY and PY), 0.70 (MY and FY), and 0.74 (PY and FY).

We also found strong positive genetic correlations between MY and PY (*r_g_* = 0.92), MY and FY (*r_g_* = 0.84), and PY and FY (*r_g_* = 0.88). In contrast, there were weak negative phenotypic and genetic correlations between MY and FP (*r**_p_* = −0.15, *r_g_* = −0.32), MY and PP (*r**_p_* = −0.20, *r_g_* = −0.44).

In this study, we used EBVs as the dependent variables for GWAS, Appendix A shows the heatmap of EBVs for five milk traits. A test-day model is used to estimate the heritability for each trait and breeding values for individuals. The heritability estimates for MY, FY, PY, FP, and PP are 0.12, 0.21, 0.23, 0.30, and 0.32, respectively (Table 1).

### 3.2. Marker Information

We conducted GWAS analyses with 1220 Holstein dairy cows and 124,743 markers after quality control (QC). Markers covered all 29 autosomes plus the X sex chromosome (Figure 1a). After the QC filtering, we re-calculated the minor allele frequency (MAF) for all SNPs. The minimum MAF was 3.8%. There were only 0.1% of markers with MAF below 5% (Figure 1c). Marker density was high. Majority of markers (56%) are within 20 kb distances to their adjacent markers (Figure 1b). Within such distance, the LD was strong (average R^2^ = 0.46) (Figure 1d).

### 3.3. Population Structure

To determine the level of population stratification, we plotted the population structure by principal component analysis (PCA). The population stratified into two unevenly sized groups (Figure 2). We also produced a scatter plot of bull’s country source (Appendix A). To adjust for the population stratification, the first three principal components (PCs) was fitted as covariate variables in the association analysis. The first three PCs explain the 1.6%, 1.3%, and 1.1% of variation, respectively, about 4% of the variation is explained by the first three PCs together. We also constructed a scatter plot between the first three PCs and the five milk traits. There are weak correlations observed between the PCs and these phenotypes (Appendix A).

### 3.4. Results of the Genome-Wide Associations

By drawing the Quantile-Quantile (QQ) plots, we found that the model for GWAS analysis in this study was reasonable, and the point at the upper right corner also shown that some significant markers were found that associated with four milk quality traits (Figure 3). We used *p* < 4.0 × 10^−7^ as the threshold, which corresponds to 1% of type I error after Bonferroni multiple test correction. A total of ten highly significant SNPs are associated with fat and protein, but no threshold significant SNP is associated with MY (Table 2, Figure 3). Three SNPs (rs42295213, rs136949224, and rs109421300) associated with FP are located on BTA1, 8, and 14, four SNPs (rs43526055, rs137676276, rs109528658, and rs135780687) associated with FY are located on BTA7, 11, 17, and X, respectively. Three SNPs (rs109875012, rs109421300, and rs108996837) associated with PP are located on BTA5, 14, and 21, respectively. One SNP associated with PY is located on BTA5. Four of these ten significant SNPs are located inside genes EPH receptor A6 (*EPHA6*), solute carrier organic anion transporter family member 1A2 (*SLCO1A2*), diacylglycerol O-acyltransferase 1 (*DGAT1*), and E1A binding protein p400 (*EP400*). The SNP (rs109875012) on BTA5 is located close to the *ZNF384* (zinc finger protein 384). The SNP (rs10705865) on BTA8 is located close to *SCARA5* (scavenger receptor class A member 5) gene. The SNP (rs137676276) on BTA11 is located close to vitrin (*VIT*). The SNP (rs108996837) on BTA21 is located close to *EXOC3L4* (exocyst complex component 3 like 4), and the SNP (rs135780687) on X chromosome is located close to *GRPR* (gastrin releasing peptide receptor). The most significant SNP (rs109421300) associated with both FP and PP is located in the *DGAT1* gene. Two SNPs (rs137676276, rs108996837) exhibit notably smaller MAFs compared to other SNPs, 0.11 and 0.12, respectively (Table 2).

### 3.5. Pleiotropic QTLs for Milk Production Traits

We used the SNPs with *p* < 0.0005 to make a heatmap to look for markers associated with two or more milk production traits because these milk traits are moderately or highly correlated (Appendix A). There are some SNPs and QTLs associated with three milk traits (MY, PY, and FY) on BTA1, 2, 3, 11, 17, 20, and 22. QTLs associated with MY while PY on BTA1, 6, 14, and 17. A QTL associated with FY and FY is on BTA20.

## 4. Discussion

### 4.1. Population Structure

Population stratification is an important confounding factor due to systematic ancestry differences that can cause false positives in GWAS [24]. By the principal component analysis, the PCA scatter plot showed that there is population structure in this studied population (Figure 2 and Appendix A). Two probably reasons for this deviation, the first reason is that those Holstein semen from overseas is not used by all cows in dairy farms, and there are still some local Holstein semen used by cows, as we know most of the farms in this study participated in a dairy breeding project that introduced Holstein semen from overseas annually from 2013 to 2018. Another reason is that some cows are introduced from different countries and contain blood from other breeds. In general, not all the registered cattle are purebreds. Especially in cattle population, this point can be explained by looking at the cattle breed registration requirements in different countries. Here, we take the Holstein cattle as an example, one of the requirements to register a Holstein cattle in China is that the cattle at least has 87.5% blood of Holstein (Chinese Holstein, GB/T 3157 2008), all the animals with Holstein genetics can be registered in Canada (https://www.holstein.ca/Public/en/Services/Registration/Registration_Eligibilities) and there are similar clauses in the USA (http://www.holsteinusa.com/animal_id/register.html). According to the above standards, we can see the common ground to register a Holstein cattle between different countries is that not all the Holstein cattle are purebreds. Even most of the Holstein cattle are purebreds, some of registered cattle could still contain a little other blood in the long-term breeding progress. That is why the population structure analysis is necessary in this study.

As we observed the population structure, the principal components were fitted as covariance to association analysis to correct population stratification. After adjusted PC factors, there are four SNPs overlapped with the association model and not fitted PCs—results will be discussed in a later section.

### 4.2. GWAS for Milk Traits

Milk production and quality are important economic objectives in the dairy industry, good milk production performance can bring greater economic benefits because of the milk pricing system and production efficiency. Most of milk phenotypes are quantitative traits and are regulated by polygenic, research on the relationship between genetics and milk traits began decades ago. As early as 1994, a research identified a QTL significantly associated with FY was linked to kappa-casein and a QTL for PY was linked to beta-lactoglobulin [25]. Subsequently, a growing number of studies detected tens and thousands of QTLs through the 30 chromosomes associated with 653 different traits in cattle (Cattle QTLdb) [26]. Even though there has been plenty of research in this filed, the available evidence is still not enough to give a completed explanation of genetic mechanism for these traits, and more new research samples will still be valuable to help lay groundwork in this aspect. Therefore, 1220 Holstein cows with comprehensive herd-test data were genotyped for GWAS in this study. In the results, we found ten SNPs were significantly associated with four milk quality traits (FP, FY, PP, and PY), one of the most significant SNPs was located in the *DGAT1* gene and shown closely related to milk fat and protein percentage. No significant SNP passed the Bonferroni correction threshold for MY, the small number of markers detected may be due to the limited sample size in the present study, which is a critical factor limiting the statistical power.

A random regression test-day model was used to estimate breeding values in this study, which eliminated environmental factors (herd-test-day, calving year-season and calving month-age), and then EBV was used as a dependent variable to conduct association tests, whereas a study used the deregressed EBV as phenotypic records for the bulls to estimate SNP effects by using a single-marker regression model [27]. The deregressed EBV was proposed by Garrick et al., in which removing the parental average effects is more valuable in genomic analysis [28]. The phenotypes used by association analysis can be varied, raw phenotypes, adjusted phenotypes, EBVs, deregressed EBVs, and daughter yield deviation (DYD) [29,30,31,32,33,34], and certainly different phenotypes are suitable for different scenarios. In dairy breeds, EBVs or deregressed EBVs are preferred as the dependent variable of GWAS. Research has shown that the deregressed EBVs could reliable in genomic analysis, but some studies showed that the accuracy of genomic analysis when using EBVs was only slightly lower than using deregressed EBVs [35,36]. In addition, a simulation study indicated that GWAS using EBVs or deregressed EBVs as the dependent variable with polygenic effect modeled had similar performance in controlling the false positive rates (FPR), even when using deregressed EBVs lowered the power in some degree [37]. Therefore, the present study prefers to use EBVs for analysis.

In this study, the three SNPs on BTA1, 8, 14 that were associated with FP were found, which are within reported QTL regions [38,39,40]. The study used Bos_taurus_UMD_3.1 as a reference genome to search the candidate genes at a distance of 120 kb upstream or downstream of the associated SNPs. The SNP (0.007%; *p* = 1.50 × 10^−7^) associated with an increase of FP on BTA1 is located in the *EPHA6* gene, which has functions of ATP binding (GO:0005524), protein binding (GO: 0005515), protein tyrosine kinase activity (GO: 0004713), and has been proposed to participate in Axon guidance pathway (KEGG: bta04360). The SNP (0.012%; *p* = 3.57 × 10^−8^) on BTA8 located close to *SCARA5* gene resulted in an increase of FP. This gene is a member of the scavenger receptor (SR) family, which is broad expression in fat tissue in humans. Research showed that scavenger receptors involved in lipid accumulation and inflammation [41]. Some studies reported the gene plays a critical role in progression and metastasis of breast cancer [42] and is involved in breast carcinogenesis [43]. The SNP on BTA14 associated both with an increase of FP (0.018%, *p* = 9.92 × 10^−25^) and PP (0.006%; *p* = 4.75 × 10^−8^), which is in the *DGAT1* gene. As we know, *DGAT1* gene is widely reported associated with milk yield and composition, especially K232A polymorphism affecting on milk fat and protein [38,44,45]. In this study, the SNP (rs109421300) we identified is within an intron of *DGAT1* gene and is the most significantly affecting FP and also affecting PP, and a lot of studies have reported for milk yield and composition in Holstein, Jersey, and Holstein–Friesian cattle [46,47,48]. *DGAT1* gene as a key metabolic enzyme catalyzes the biosynthesis of triacylglycerols [49], and glycerolipid metabolism (KEGG: bta00561), retinol metabolism (KEGG: bta00830), metabolic pathways (KEGG: bta01100), and fat digestion and absorption (KEGG: bta04975) by KEGG pathway analysis (https://www.genome.jp/dbget-bin/www_bget?bta:282609). The SNP (−0.004%; *p* = 4.03 × 10^−8^) on BTA5 is associated with a reduction of PP and is within previously reported milk fatty acid content QTL [39]. This SNP is close to the *ZNF384*, which encodes a C2H2-type zinc finger protein. It may function as a transcription factor and have DNA binding and metal ion binding functions. Another SNP (−0.008%; *p* = 2.36 × 10^−8^) associated with a reduction of PP on BTA21 at 69Mb is located near the *EXOC3L4* gene. Four SNPs are associated with FY in the present study. The SNP (−1.523 kg; *p* = 4.48 × 10^−9^) on BTA7 near the adrenoceptor alpha 1B (*ADRA1B*) gene resulted in a reduction of FY, which is participated in the calcium signaling pathway (KEGG: bta04020), cGMP-PKG signaling pathway (KEGG: bta04022), and neuroactive ligand–receptor interaction (KEGG: bta04080). The SNP (−2.281 kg; *p* = 8.58 × 10^−9^) associated with a reduction of FY on BTA11 at 19Mb located near the VIT gene. This gene encodes an extracellular matrix (ECM) protein and has been suggested to contribute to normal brain asymmetry variation [50]. The SNP (1.543 kg; *p* = 7.05 × 10^−9^) on BTA17 is located in the *EP400* and resulted in an increase of FY, and also within previously reported QTL for milk protein composition [51], *EP400* gene participates in histone H2A acetylation, histone H4 acetylation, and has the function of ATP binding, DNA binding, protein binding, and helicase activity. The SNP (1.629 kg; *p* = 1.63 × 10^−10^) associated with an increase of FY on chromosome X is located nearby the *GRPR*. This gene is a gastrin releasing peptide receptor. Gastrin-releasing peptide regulates numerous functions of the gastrointestinal and central nervous system, which is involved in calcium signaling pathway (KEGG: bta04020) and neuroactive ligand–receptor interaction (KEGG: bta04080). These two pathways were enriched for FY in this study. A study revealed that the calcium signaling pathway was related to milk coagulation properties and curd nutrient recovery traits in dairy cattle [52], and interestingly, another study showed that these two pathways were significantly correlated with lactation performance in mice [53]. Song et al. reviewed a lot of studies on calcium signaling pathway that participated in the effect of the sympathetic nerve in regulating adipose metabolism [54]. We suspect that *ADRA1B* and *GRPR* genes may affect milk fat through the calcium signaling and neuroactive ligand–receptor interaction pathways. It is well known that the biological mechanisms of quantitative traits are very complicated, and the present study is only based on SNP data analysis. Some studies have shown that copy number variation and DNA methylation are also related to milk production and quality traits [55,56,57]. The SNP (1.192 kg; *p* = 1.57 × 10^−8^) on BTA5 is associated with an increase of PY and it is also within the previously reported QTL region, which is associated with milk fatty acid content. This SNP is in the *SLCO1A2* gene, which is encoding solute carrier anion transporter family, member 1A2, the gene participating in the digestive system, organic anion transport process, and bile secretion pathway (KEGG: 04976). Furthermore, when the association model used not fitted PCs as covariance, four SNPs are overlapped with the model with fitted PCs: rs109875012 (*ZNF384*), rs136949224 (*SCARA5*), rs109421300 (*DGAT1*), and rs109528658 (*EP400*), it is suggested that these significant SNPs could be more stable and reliable.

### 4.3. Correlations among Milk Traits and Pleiotropic QTLs

Estimates of heritability and genetic correlations are essential population genetic parameters in animal breeding research and application of animal breeding programs [58]. Genetic correlation can be useful for indirect selection, selection in different environment, and selecting multiple traits simultaneously. We found there was relatively high genetic correlation between milk yield and protein yield, and it is consistent with the results in Holsteins [59] and Jersey cattle [58]. We expect that there are overlapping regions of genetic variation between different traits that are relatively high correlation. A heatmap using *p*-values from GWAS results helped identify pleiotropic QTLs (Appendix A). The QTL on BTA2 at 134Mb is associated with MY, PY, and FY, and also reported associated with the milk composition in another Holstein study [60]. The MY- and PY-associated QTL is adjacent to the *ABCG2* gene on BTA6; previously, studies reported that a missense mutation in the *ABCG2* gene is associated with milk yield and composition in Holstein, Braunvieh, and Fleckvieh cattle [2,61], an intron variant affecting milk fatty acids in Chinese Holstein [62], and has also been reported to affect body weight, calving ease direct in US cattle breed [63,64]. The QTL on BTA17 is associated with MY, PY, and FY, with previous studies indicating that QTLs in this region are related to milk fatty acid and body weight [60,65]. A previous study reported a QTL on BTA20 is associated with PP and body weight [63,66]. In this study, we found this QTL is associated with MY, PY, and FY. The QTL on BTA24 is pleiotropic and is associated with MY and FY, and previously reported for milk fatty acid [67] and conception rate in Holstein cows [68].

## 5. Conclusions

The study performed genome-wide association analysis using test-day records for five milk production and quality traits in Holstein cows. A total of ten significant SNPs associated with milk fat and protein percentage, fat and protein yield were found, six of them located within previously reported QTLs, including *DGAT1* gene on milk quality. We also found some new genetic loci and candidate genes related to milk fat and protein. In addition, some SNPs and QTLs associated with more than two milk traits were found. These results could provide some basis for molecular breeding and useful information to understanding the genetic architecture of milk production and quality traits in dairy cattle.

## Figures and Tables

**Figure 1 animals-10-02048-f001:**
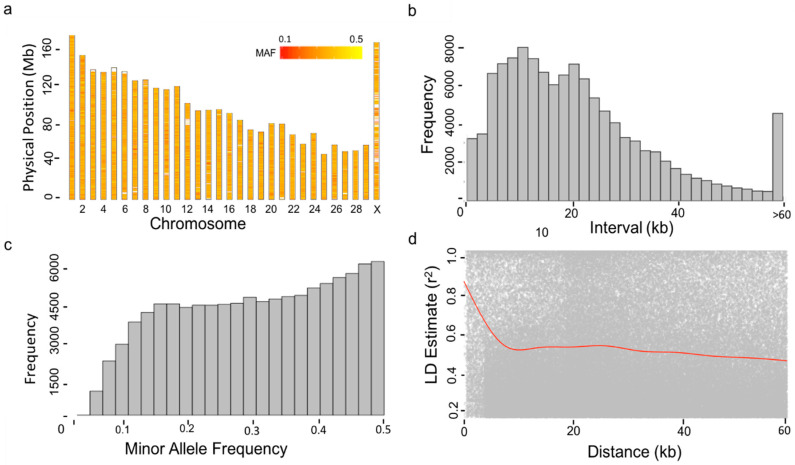
Properties of Single Nucleotide Polymorphisms (SNPs). A total of 1220 cows were genotyped by the Illumina Bovine 150 k BeadChip. After conducting quality control on both minor allele frequency (MAF, above 5%) and missing rate (<10%), 1220 individuals and 124,743 SNPs remained. The distribution of the filtered SNPs is displayed over the 30 bovine chromosomes except for Y chromosome (**a**). The MAFs of SNPs were re-calculated after the filtering and were displayed by a heat map. Consequently, the SNPs with MAF < 5% remained, as demonstrated by the histogram (**c**). The density of SNPs is displayed by the frequency of the distance between adjacent SNPs (**b**). The distances over 60 kb clustered into one group. The maximum distance was 100.06 kb. Pairwise Linkage Disequilibrium (LD) was calculated as the R square for SNPs within the 100 kb window. The decay of LD over distance (red line) is displayed by the pairwise LD and moving average (**d**).

**Figure 2 animals-10-02048-f002:**
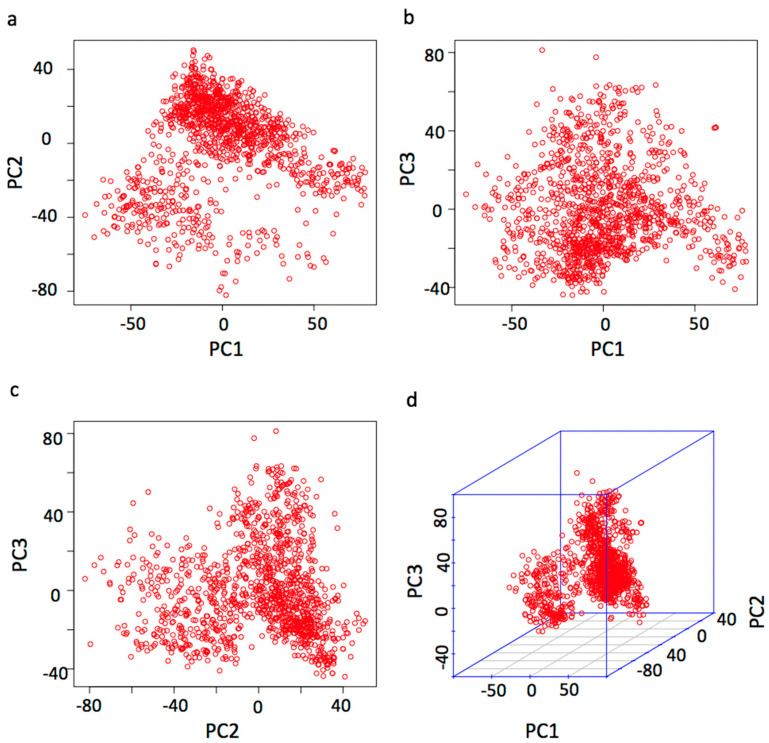
Population structure demonstrated by principal component analysis. Principal component analysis (PCA) was conducted with the 124,743 SNPs for the 1220 cows. The population structure is demonstrated by the pairwise scatter plots (**a**–**c**) and the 3D plot (**d**) of the first three principal components (PCs).

**Figure 3 animals-10-02048-f003:**
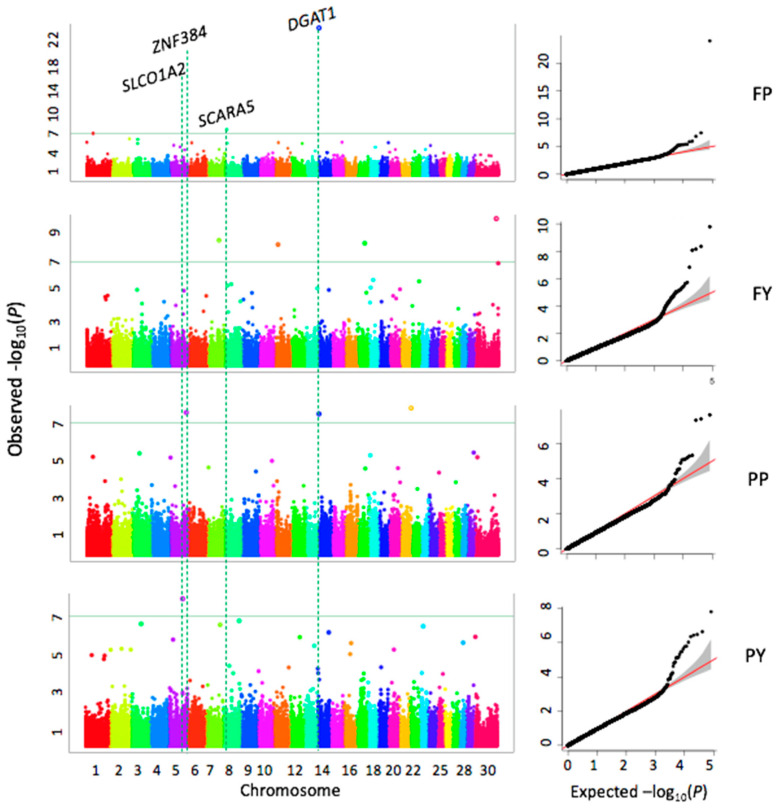
Associations between 124,743 SNPs and milk traits. Milk traits include fat yield (FY), protein yield (PY), fat percentage (FP), protein percentage (PP). The association analyses were conducted by the FarmCPU R package. Manhattan plots display the negative logarithms of the observed *p* values for SNPs across 30 bovine chromosomes (left panel). The green line indicates the Bonferroni multiple test threshold at *p* = 4.0 × 10^−7^. The Quantile-Quantile (QQ) plots represent the negative logarithms of the expected *p* values (*X*-axis) and observed *p*-values (*Y*-axis) (right panel).

**Table 1 animals-10-02048-t001:** Variance components and heritability of milk traits *.

Variance Component	MY	FP	FY	PP	PY
Genetic	1592.62	4.76	1.48	1.65	1.20
Permanent environmental	4267.93	10.58	5.46	3.45	3.96
Residual	7413.63	0.49	0.08	0.07	0.03
h^2^(SE)	0.12(0.01)	0.30 (0.05)	0.21(0.02)	0.32(0.01)	0.23(0.02)

* MY, milk yield; FP, fat percentage; FY, fat yield; PP, protein percentage, PY, protein yield; SE, standard error.

**Table 2 animals-10-02048-t002:** Genome-wide significant SNPs associated with milk traits *.

Traits	SNP	CHR	Position (bp)	MAF	Nearest Gene	Distance (kb)	*p-*Value	Effect
FP	rs42295213	1	41,061,715	0.36	*EPHA6*	within	1.50 × 10^−7^	0.007
PP	rs109875012	5	104,120,905	0.45	*ZNF384*	2.6	4.03 × 10^−8^	−0.004
PY	rs134480235	5	89,267,320	0.49	*SLCO1A2*	within	1.57 × 10^−8^	1.192
FY	rs43526055	7	73,431,219	0.31	*ADRA1B*	179.8	4.48 × 10^−9^	−1.523
FP	rs136949224	8	10,705,865	0.14	*SCARA5*	43.5	3.57 × 10^−8^	0.012
FY	rs137676276	11	19,277,448	0.11	*VIT*	25	8.58 × 10^−9^	−2.281
FP PP	rs109421300	14	1,801,116	0.23	*DGAT1*	within	9.92 × 10^−25^ 4.75 × 10^−8^	0.018 0.006
FY	rs109528658	17	46,090,458	0.40	*EP400*	within	7.05 × 10^−9^	1.543
PP	rs108996837	21	69,386,346	0.12	*EXOC3L4*	32.6	2.36 × 10^−8^	−0.008
FY	rs135780687	X	134,726,985	0.42	*GRPR*	83	1.63 × 10^−10^	1.629

* FP, fat percentage (%); PP, protein percentage (%); PY, protein yield (kg); FY, fat yield (kg).

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
