# Peer review of "GWAS-Based Identification of New Loci for Milk Yield, Fat, and Protein in Holstein Cattle"

_animals, 2020, doi:10.3390/ani10112048_

Round 1
Reviewer 1 Report
All comments, suggestions and corrections were now considered in this revise version.
I juts have a comment with respect to a reference, which is give in Chinese. I recommend a full translation for English. Follow bellow the mentioned reference:
1. REN X L. Establishment of genetic evaluation system for lactation performance of Chinese Holstein cattle in Ningxia (宁夏地区中国荷斯坦牛泌乳性能遗传评估系统的建立)[D]. Bei jing: China Agricultural University (北京: 中国农业大学), 2015.
Author Response
Thank you for your comment.
This reference has been full translated for English, please check reference 18 in revision.
Reviewer 2 Report
Line 182: Correct "was corresponds" to read "corresponds"
Author Response
Thank you for your comment.
We have corrected "was corresponds" to read "corresponds". Please check the revision.
Reviewer 3 Report
To Authorsdocument animals-958618 describe SNP effects on classical traits in dairy cattle.
The Association analysis is not clear and some more information are needed in the M&M section.
Try to comparare your approch in the discussion respect to approach as Viale et al. 2017,
J. Dairy Sci. 100:7271–7281,https://doi.org/10.3168/jds.2017-12666,
where they used de-regressed EBV to estimate the SNP effects.
Author Response
Describe SNP effects on classical trairs in dairy cattle.
Thank you for your comment. We added description of SNP effects on milk traits in discussion section. Please check the revision.
The Association analysis is not clear and some more information are needed in the M&M section.
Thank you for your comment. We added association test method and description of models in the M&M section. Please check the revision.
Try to comparare your approch in the discussion respect to approach as Viale et al. 2017,
J. Dairy Sci. 100:7271–7281,https://doi.org/10.3168/jds.2017-12666,
where they used de-regressed EBV to estimate the SNP effects.
Thank you for your comment. We added some information and compared the approach of my study in the discussion respect to approach as Viale et al. 2017. Please check the revision.
In addition, we added some more information in the first paragraph of the introduction and the second part of the discussion. Please check the revision.
Round 2
Reviewer 3 Report
To Authors
paper is improved but the Association analysis need to be improved because of the direct use of the EBVs it is not the best way. EBVs have differnete accuracy so the results it is not reliable. We suggest to apply , as y-dependent variable the de-regressed proofs of EBVS as reported by Viale et al. (2017) J. Dairy Sci. 100:7271–7281 https://doi.org/10.3168/jds.2017-1266.
Round 3
Reviewer 3 Report
Paper version now is mature to be published.
This manuscript is a resubmission of an earlier submission. The following is a list of the peer review reports and author responses from that submission.
Round 1
Reviewer 1 Report
This is an important study that contributes to our understanding of the genetic underpinnings of milk production and quality traits.
To help improve on the quality of the manuscript, I have made all suggestions in text as shown in the attached edited manuscript. However, a major challenge of the manuscript is flow of ideas and communicating to the reader. Authors should get a someone to proof-read the final draft and correct all grammatical errors before submission.
The authors are also invited to correct "principle" to read "principal" in Figure S3 of the supplementary material.
Thank you.

Reviewer 2 Report
Dear Authors,
Although the manuscript is interesting and quite original, I think it needs some revisions to improve it. In particular, I suggest to improve the "discussion" section which seems very poor.
lines 24-25: please avoid repetitions (yield, percentage)
line 30: fixed instead of "Fixed"
lines 36: in my opinion keywords should be changed
lines 51-53: improve these sentences
lines 54-55: the sentence seems incomplete
line 57: please avoid to repeat the term "milk"
lines 61-63: the meaning of this sentence is not so clear...
line 65: mutations instead of "mutation"
lines 66-67: in my opinion, reference to maize selection is not appropriate
line 69: "shrimp breed" - are you sure the term breed is appropriate?
lines 71-76: improve the aim of the study
line 79: studied instead of "study"
line 82: "and the start of lactation" ?????
lines 90-92: improve this stentence
lines 94-95: "fixed regression...." the sentence seems incomplete
lines 95-99: please verify subscripts of model equation and add levels for fixed effects
line 109: the sentence is not clear
line 112: MAF spell it at first use
lines 128-129: principal component analysis
line 131: "The of origins of the sires were" what does it means? please correct
line 184-185: improve this sentence
line 209-216: please re-word this paragraph
line 221: "full blood" is not the right term
lines 233-236: improve this sentence
lines 238-240: improve the text
lines 245-245: this statement needs some references
lines 250-251: this sentence is not clear
line 271: "Another PP-associated SNP on BTA21" ....????
lines 298-299: improve this sentence
lines 314-322: re-word the conclusion seection
Reviewer 3 Report
GENERAL COMMENTS
The manuscript is interesting; however, many issues related to statistical analysis and modeling must be elucidated. Detailed descriptions of the main points to be improved are described below:
TITLE
“Ten Genetic Loci Identified for Milk Yield, Fat, and Protein in Holstein Cattle”
This tittle is quite general and few attractive for the readers. I suggest including information on “genome-wide association analysis” as well as the “several new candidate genes” identified in this manuscript (see SIMPLE SUMMARY section). I’m only suggesting a more attractive title according to the manuscript content as follow: “GWAS-based Identification of New Loci for Milk Yield, Fat, and Protein in Holstein Cattle”
ABSTRACT
Lines 27-28: “using a mixed linear model for individuals with and without phenotypic data.”
The correct description is “linear mixed model”. Furthermore, there are no reasons to describe on “for individuals with and without phenotypic data”, since this advantages is very clear in the “linear mixed model” theory.
Line 31: “above the genome-wide significant threshold”
The reader must to known the used “threshold” to identify a significant marker. This information must be briefly reported here.
Lines 33-35: “The most significant SNP is on DGAT1 gene affecting milk fat and protein percentage. These genetic variants and candidate genes would be valuable resources to enhance dairy cattle breeding.”
Based on these results, what are the innovations of the present manuscript? At first, there are evidences on “new results” in this study. Additionally, how to exploit these results “to enhance dairy cattle breeding.”? A briefly comment about this is required here.
INTRODUCTION
Lines 41-42: “With the advances in molecular genetic techniques, genomic selection (GS) has been widely used in plant and animal breeding.”
The present manuscript is not related to genomic selection (GS), but with GWAS (genome-wide association study). This sentence must be revise din the context of this manuscript.
Lines 44-45: “and the method of integrating Genome-wide association studies (GWAS) and GS into one step has been developed, which account the genetic architecture of interest traits”
This kind of “integration” was not used here. The authors must to rewrite the INTRODUCTION section focusing in the reported results.
Lines 45-47: “The conventional genomic best linear unbiased prediction (GBLUP) considering all markers that have the same effects, so thus ignores the markers that have large effect on target traits.”
I cannot see reasons to include this information over here. Why to make mention on GBLUP under a GWAs context? This sentence must be revised.
Lines 48-49: “Research showed that the prediction accuracy was higher than that use of conventional GBLUP when fitting the most significant marker in the prediction model”
The prediction accuracy was not the aim of this manuscript. This sentence must be revised. In summary, this paragraph would be revisited by focusing in the real aims of the present manuscript.
Line 69: “marker density in cattle breed[17] and shrimp breed[18]”
There is no reason to report previous results obtained from other species, such as “shrimp breeding”. This citation would be removed.
Lines 69-70: “Therefore, it is necessary to use dense genotype to identify genetic variation and implementation of genomic prediction”
The present manuscript it is not related to “genomic prediction”. This issue has been already previously highlighted (please see lines 41-42).
Lines 75-76: “would become valuable resources for genomic evaluation.”
What means “genomic evaluation” in the context of the present manuscript? It is not clear. Please, see several comments about this question.
MATERIAL AND METHODS
Lines 90-91: “Totally about 452,920 test-day records from 61,600 cows spanning a 9-year period (2011-2019) at their first lactation stage”
This kind of information needs to be reported at the previous sub-topic entitled “2.1. Population and phenotypic data”. It is because the present information treats on “phenotypic information”.
Lines 94-95: “Fixed regression using a fourth order Legendre polynomial”
What is the justification to use this polynomial order without previous comparison with other ones? How to know if this order is suitable to describe the data assumed in the present study?
Lines 96-105: Statistical properties of the covariance matrices for model coefficients as well as for the residual variance components must be reported here. Additionally, what about the heterogeneity of residual variance? Have the authors tested different models approaching this issue? More information is requited here. Furthermore, information on the used “genomic relationship” matrix is essential for this kind of test-day modeling. Why there is no information on this matrix property?
Line 107: “Blood samples from the 1,220 cows were collected by cattle farm staff in this study”
This is a very small sample to be considered in genomic analysis of dairy cattle. Which are the justifications to use this small sample size in the present study? What about “reliable inferences” in the presence of small number of genotyped animals? More information is required here.
Line 128: “2.4. Principle component analysis”
The name of this method is wrong. The correct name is “Principal component analysis”
Lines 131-132: “The of origins of the sires were plotted against the principal components to demonstrate how the population structure was formed.”
Firstly, previous citations approaching PC analysis “to demonstrate how the population structure was formed” based on the origins of the sires are required here. If there are no previous references, the authors must to explain better about this strategy. Secondly, what is the variation percentage explained for the used number of PC? (Please, see line 139 “first three principal components as covariate variables in the GWAS models.”).
Line 142: “genes were found within distances of 120 kb upstream or downstream to the associated SNPs.”
Why 120 kb? Are there previous references? If not, have the authors used the LD decay graph (Figure 1 d) to infer on this distance? Explanations are required over here.
RESULTS
Table 1. Variance components and heritability of milk traits
Some kind of “uncertainty measures” such as standard errors (SE) must be provided in this Table. There is no way to infer on genetic parameters without take into account “uncertainty measures”. In summary, the SE are required in this Table.
Lines 152-153: The standard errors for genetic and phenotypic correlations are also required over here. How to infer on these correlation without consider “uncertainty measures”? This is a kind of basic information widely required in scientific publications in Genetics and Animal Breeding.
DISCUSSION
The discussion depends on the justifications and/or corrections reported at MATERIAL AND METHODS section.
